# Targeting NF-κB Signaling in Cancer Stem Cells: A Narrative Review

**DOI:** 10.3390/biomedicines10020261

**Published:** 2022-01-25

**Authors:** Barbara Kaltschmidt, Kaya E. Witte, Johannes F. W. Greiner, Florian Weissinger, Christian Kaltschmidt

**Affiliations:** 1Molecular Neurobiology, Faculty of Biology, Bielefeld University, Universitätsstrasse 25, 33615 Bielefeld, Germany; barbara.kaltschmidt@uni-bielefeld.de; 2Forschungsverbund BioMedizin Bielefeld FBMB e.V., Maraweg 21, 33617 Bielefeld, Germany; kaya.witte@uni-bielefeld.de (K.E.W.); johannes.greiner@uni-bielefeld.de (J.F.W.G.); Florian.Weissinger@evkb.de (F.W.); 3Department of Cell Biology, Bielefeld University, Universitätsstrasse 25, 33615 Bielefeld, Germany; 4Department of Hematology, Oncology, Internal Medicine, Bone Marrow and Stem Cell Transplantation, Palliative Medicine, and Tumor Center, Protestant Hospital of Bethel Foundation, University Hospital OWL of Bielefeld University, Schildescher Str. 99, 33611 Bielefeld, Germany

**Keywords:** cancer stem cells, NF-κB, EGF, PD-L1, lenalidomide, bortezomib, dexamethasone

## Abstract

Among the cell populations existing within a tumor, cancer stem cells are responsible for metastasis formation and chemotherapeutic resistance. In the present review, we focus on the transcription factor NF-κB, which is present in every cell type including cancer stem cells. NF-κB is involved in pro-tumor inflammation by its target gene interleukin 1 (IL1) and can be activated by a feed-forward loop in an IL1-dependent manner. Here, we summarize current strategies targeting NF-κB by chemicals and biologicals within an integrated cancer therapy. Specifically, we start with a tyrosine kinase inhibitor targeting epidermal growth factor (EGF)-receptor-mediated phosphorylation. Furthermore, we summarize current strategies of multiple myeloma treatment involving lenalidomide, bortezomib, and dexamethasone as potential NF-κB inhibitors. Finally, we discuss programmed death-ligand 1 (PD-L1) as an NF-κB target gene and its role in checkpoint therapy. We conclude, that NF-κB inhibition by specific inhibitors of IκB kinase was of no clinical use but inhibition of upstream and downstream targets with drugs or biologicals might be a fruitful way to treat cancer stem cells.

## 1. Introduction into Scope of Review

Here, we briefly discuss the scope of the review and our search strategy. Nuclear factor ‘kappa-light-chain-enhancer’ of activated B-cells (NF-κB) is a transcription factor complex active in every cell type including cancer stem cells but not in embryonic cells [1]. The signaling pathways of NF-κB in cancer stem cells (CSCs) were the subject of a recent review [2]. Thus, we thought a current review on targeting NF-κB by chemicals and biologicals within an integrated cancer therapy with relevance to this important cell type might be of interest to a biomedical audience. We used an advanced pubmed.gov search strategy with MeSH terms cancer stem cell AND adjuvant drug therapy, we got 52 hits, but none with NF-κB. Therefore, we made more focused searches on known components of NF-κB signaling. Furthermore, we briefly included historicity that was appropriate. For the selection of approaches in integrated therapy, we used experience gathered within the local clinic together with current german guidelines in onkopedia (https://www.onkopedia-guidelines.info/en/onkopedia/guidelines, accessed on 23 December 2021).

## 2. Cancer and Cancer Stem Cells

### 2.1. Development of Cancer Stem Cells and Cancer Origin

Cancer is commonly known as a leading cause of death worldwide with an estimated 19.3 million new cases per year [3]. Generally, the term tumor by including cancer is defined by the abnormal growth of different cell types, which thus combines a huge group of diseases. State-of-the-art classification initially divides those tumoral diseases into benign and malignant forms. While benign tumors were explained as non-cancerous and unable to metastasize, the malignant growth of a tumor is characterized by the infiltration of surrounding tissue, the potential to spread in patients’ bodies via blood- and lymph system for metastasis as well as the existence of nearly immortal tumor cells. Therefore, exclusively malignant tumors are designated as cancer, and the related subgroup of immortal cells are regularly mentioned as cancer stem cells. These cancerogenic cells with stemness characteristics were broadly described as the major reason for cancer, which emphasizes the studying of cancers’ initiation, progression, and recurrence as an essential field in current medical- and biological research. Firstly introduced in 2001, CSCs were reported in comparison to hematopoietic stem cells (HSCs), sharing useful properties and signaling pathways [4]. Thus, Reya and colleagues postulated tumors and respective CSCs to mainly arise from normal stem cells. From a recent point of view, the development of cancer in direction of CSCs appears wide more complex, whereby various mechanisms such as cellular de-differentiation processes or the influence on burden organs and affected tissues take a center stage. On this basis, many studies focused on the research of CSCs within the human body and the resulting impact on cancers’ micro- and macroenvironment.

### 2.2. Micro- and Macroenvironment of Cancers

Emphasizing the influence on cancers’ localization and respective surrounding, the macro- and microenvironments of a tumor were reported as crucial for cancer spreading and development. In particular, the microenvironment of a tumor is known as localized proliferation-promoting auto- and paracrine interactions of cancer cells to their surrounding tissue [5], reviewed in [6,7]. Here, specific types of cancerogenic cells can evade patients’ immune system within the close tumor proximity, release soluble factors, like cytokines or growth factors and upregulate cancer cell division or induce necrosis in the direct tumor surrounding. Thus, different cancer-associated mechanisms are initiated such as the systemic nutrient supply via tumoral recruitment and reconstitution of blood vessels, named angiogenesis [8]. Moreover, systemic dysfunction and reshaping of the immune composition of patients could be observed in different tumor models [9]. Explaining the resulting tumor macroenvironment, Al-Zhoughbi and co-workers reviewed the systemic pathological interaction between various cancer- and normal cells compared to the previously localized interactions in the microenvironment of tumors [7]. Thus, special consideration of the cellular structure within the malignant tumor building is mandatory for a detailed understanding of the prevalent cancer-mediated mechanisms and interactions.

### 2.3. Cellular Hierarchy and Cancer Stem Cell-Specific Characteristics

Concerning the biological cancer composition, analysis of the characteristic cellular structure takes a center stage in current bio-oncological research. Focusing on the universal cellular tumor structure, two main models are commonly discussed. As recapitulated by Vermeulen and Sousa e Melo in 2012, the original model described CSCs at the top, which give rise to progenitor-like cells and further to differentiated tumor cells [10]. This hierarchically organized model was for a long time assumed as the sole prevalent structure of tumors, based on the above-mentioned hypothesis of CSCs exclusively originating from normal stem cells like HSCs [4]. From a recent perspective, the CSC phenotype is also known to be under the distinct adjustment of the cellular environment and thus displays an emerged dynamic between differentiated stages, progenitor-like cells, and CSCs (reviewed in [10]). In this line, the procedure of de-differentiation was shown by various studies describing the rise from differentiated cancer cells up to CSCs, directed by the microenvironment of the tumor [11,12,13]. Here, especially stromal cells directly surrounding the tumor-like mesenchymal stem cells, infiltrating immune cells, myofibroblasts, or endothelial cells are related to process CSC-dependent signals, which are also under the control of cancer cells themselves. Consequently, the mutual interference of cancer- and stromal cells within the tumor-infiltrating zone seems to display a central role in cancer progression and proliferation. Nevertheless, CSCs are defined with some additional benefits in contrast to differentiated cancer cells, which are currently presented as unique characteristics (Figure 1, magenta features).

Firstly, designated in context to the original cellular cancer hierarchy, the self-renewing capacity of CSCs is an essential quality of their stemness character (reviewed in [14,15]). Next to self-renewal, the tumorigenic properties including multi-lineage differentiation and resistance to chemotherapeutics are further key features of CSCs [16,17,18]. Comprehensive investigations of these crucial CSC-associated capacities revealed the presence of a distinct expression profile, inherent genes entitled as CSC markers. In particular, the expression of surface biomarkers CD44, prominin-1 (CD133), as well as Nestin, were frequently reviewed [19,20,21,22,23,24,25], next to the expression of aldehyde dehydrogenase 1 (ALDH1), which indicates chemoresistance and poor prognosis of e.g., prostate cancer patients [26]. As a further marker, the epithelial cell adhesion molecule (EPCAM) was linked directly to the epithelial-to mesenchymal-transition (EMT), another CSC-associated and therapy-impeding mechanism [27]. Besides exclusive CSC-related characteristics, (partially) differentiated cancer cells share important features with CSCs, which are also known to reduce patients’ clinical outcomes (Figure 1, black characteristics). One of these capabilities is the already stated mechanism of tumor vascularization, whereas the nourishment of cancer is assured by the reconstitution, recruitment, or tapping of a patient’s blood vessels in close tumor proximity (reviewed in [28]. While vascular mimicry is progressed by cancer cells forming vessel-like structures, CSCs use their trans-differentiation potential to become endothelial-like cancer cells and ensure the neo-vascularization via angiogenesis. Thus, the nutritional supply of tumors displays the main part in carcinogenesis, whereas in this context autophagic regulation is also essential due to the hypoxic milieu and respective importance of maintaining cellular homeostasis in tumoral buildings (recapitulated in [29]). Regarding the further maintenance of CSCs and differentiated cancer cells, another cancer-associated characteristic is immune checkpoint inhibition, which is currently discussed in the literature as well as in relation to patient-specific therapies. Here, genes like the programmed cell death ligands 1 and 2 (PD-L1/-2) or the major histocompatibility complex class I (MHC class I) were related to the specific inhibition of patient’s immune system, whereas cancer cells equipped with these surface profiles imitate regular immune cells and subsequently evade immunosurveillance programs [30]. Next to all the above-mentioned features, a key characteristic of cancer is the regulation of the inflammatory response is, additionally linked to each referenced cancer-mediated benefit. Stated as a crucial player of inflammation, the the transcription factor NF-κB is strongly correlated to chronical diseases and cancers (reviewed in [31]). In this light, Greten and colleagues as well as Xia and coworkers associated the specific deregulation of NF-κB with tumor progression and formation [32] (also reviewed in [33]). Furthermore, the variances and implications of NF-κB signaling in organ-specific cancer were recently recapitulated [2] and comprehensively explained the impact of heterogeneity in tumors for potential targeting and therapy of CSCs from different origins.

In this line, we could recently establish various CSC lines from non-small cell lung cancer, prostate cancer, colon cancer, glioblastoma, and endometrial cancer. All CSC lines from these diverse tumors of origin shared the mRNA expression of CD44 [23,25]. CSC growth in vitro could be efficiently suppressed by pharmacologically inhibiting NF-κB with the antioxidant pyrrolidine dithiocarbamate (PDTC) and inhibition of MYC/MAX signaling with the small molecule KJ-Pyr-9. Surprisingly, no synergistic effect on cell death was observed when both NF-κB and MYC pathways were inhibited [22]. Prostatic CSCs could be efficiently killed by TNF-alpha when cell lines had a blunted NF-κB pathway [24].

### 2.4. Influence of Tumors’ Heterogeneity on Therapy of Cancer Stem Cells

Inter- and intratumoral heterogeneity are described as patient-specific variances of cancers (intertumoral) or within a single tumor (intratumoral). The cellular subgroup of CSCs was also contributed to both categories of tumors’ heterogeneity (reviewed in [34]). Firstly, regarding the intertumoral heterogeneity, intrinsic diversities between cancer patients within the same organ are well known and thought to be a major reason for poor overall survival. In this line, many studies reported distinct variations of similar tumor types of different patients [35,36,37]. While histopathological and oncological classification showed equal tumor grading, genetic characterization revealed a strong molecular heterogeneity with variances in the mutational analysis as well as in copy number alterations of different genes [35]. Focusing further particularly on a single tumor, the aforementioned cellular hierarchy, extrinsic factors like hypoxia, the microenvironment of a tumor or inflammation as well as intrinsic factors such as the genetic and epigenetic impact influence the intratumoral heterogeneity [38]. Outlined from a CSC perspective by Prasetyanti and Medema, the resulting heterogenetic impact comprises alterations of the cellular growth rate and apoptosis, genetic abnormality as well as phenotypic changes via modulation of cellular surface marker expressions (reviewed in [34]). Hardly limiting the prospects of unifying treatment strategies, inter- and intratumoral heterogeneity is thus of critical importance in cancer therapy.

## 3. The Transcription Factor NF-κB as a Target in Cancer Stem Cells

### 3.1. Introduction to Alternative and Classical NF-κB Signaling

The Nobel Prize laureate David Baltimore first described NF-κB as a dormant transcription factor, in which binding activity can be induced [39]. The dormant state of NF-κB was explained later by Baeuerle and Baltimore with the interaction of NF-κB and IκB (inhibitor of κB) in the cytoplasm, [40]. In humans, the NF-κB family comprises five members with DNA-binding ability. Among these, REL (c-REL), RELA (p65), and RELB (RELB) have transactivation domains, while NFKB1 (p50) and NFKB2 (p52) lack these domains. In classical NF-κB signaling, a DNA-binding NF-κB dimer, e.g., RELA (p65) or NFKB1 (p50) is held in its inactive state via the interaction with IκB (Figure 2) [41]. In particular, binding of IκB to p65/p50 results in an alpha-helical conformation of the p65 nuclear translocation signal sequence (NLS), in turn hindering its nuclear translocation [42]. On the contrary, a disordered NLS sequence allows the translocation of p65/p50 into the nucleus upon stimulation of NF-κB signaling (see below). Next to NF-κB, IκB is likewise able to undergo nuclear translocation, suggesting an interaction with DNA-bound NF-κB, which may, in turn, result in the induction of nuclear exports [43]. There are at least two common pathways activating NF-κB: the alternative (non-canonical) and the classical (canonical pathway (Figure 2). In the alternative pathway, the binding of ligands to lymphotoxin beta receptor or BAFF receptor activate NIK (NF-κB inducible kinase), which in turn activates a dimer of IκB kinase 1 (IKK-1) (Figure 2). 

Thereafter, activated NIK mediates the phosphorylation of IKK1, which phosphorylates the p52-precursor p100 and allows nuclear translocation of p52/RELB followed by target gene expression. The transcriptional active dimer composed of DNA-binding subunits p52 and RELB is mostly involved in lymphoid organogenesis [31].

The classical (canonical) pathway involves the IκB family. As introduced above, p65/p50 are kept in an inactive state in the cytoplasm due to complex formation with IκB family members (e.g., NFKBIA; Figure 2, classical pathway)(reviewed in [44]). To date, more than 300 proteins possessing an IκB activity are described in the literature [45]. Classical NF-κB signaling is driven by a large number of stimuli comprising amongst others inflammatory cytokines, bacterial lipopolysaccharides, viral RNA, or even neurotransmitters [46,47]. By way of example, binding of inflammatory cytokines (e.g., TNFα) to their respective receptors (e.g., TNFR1) results in IKK-dependent phosphorylation and proteasomal degradation of IκB, which in turn leads to the nuclear import of p65/p50 as detailed above (Figure 2). Binding of nuclear NF-κB to its target gene promoters’ initiates gene expression of various target genes (see T. Gilmore’s webpage https://www.bu.edu/nf-kb/gene-resources/target-genes/, accessed on 20 December 2021). Classical and/or alternative signaling of NF-κB is vital for regulating a broad range of cellular processes including, inflammation, proliferation, but also inflammatory diseases and cancer (see [31,47,48,49] for a detailed review).

### 3.2. Systemic Therapy Targeting NF-κB

The German medical scientist Paul Ehrlich (Nobel prize 1908) developed chemotherapy, which he defined as a treatment of diseases and parasites with chemical substances and a theory on the function of antibodies [50]. In the tradition of Ehrlich’s magic bullet, specific chemotherapy with small molecule drugs was developed for kinases involved in cancer cell signaling. Initially, the efforts for the development of such drugs were hampered by the high concentration of cellular ATP and the un-completely understood regulation of kinase activity, with its conserved ATP-binding pocket. In our days, about 98 small molecule drugs inhibiting kinases were approved by the FDA, but none for the serine-specific kinase complex IKK [51]. In this line, small molecules inhibit the EGF receptor as the receptor tyrosine kinase inhibitor starting with erlotinib and several others such gefitinib and afatinib up to the third generation of tyrosine kinase inhibitors: osimertinib and others could inhibit NF-κB activation by the EGFR (Figure 3). However, inhibition of the EGFR oncogene could induce an EGFR-TRAF2-RIP1-IKK complex in turn stimulating cell survival in an NF-κB-dependent manner in cultivated cells [52]. Accordingly, glioblastoma patients with NFKBIA deletion show reduced survival rates, as shown by Bredel and colleagues [53]. In particular, the 790 human glioblastomas analyzed by Bredel and coworkers revealed either an EGFR amplification or an NFKBIA deletion, with heterozygous deletions of NFKBIA being present in 28% of glioblastomas and in 22% of cancer stem-like cells. These findings strongly indicate a mutual exclusion of either amplification of the EGFR oncogene or deletion of NFKBIA. These seem to constitute two groups with NF-κB activation by two different mechanisms: either EGFR amplification or deletion of NFKBIA.

A milestone for potential targeting NF-κB with its specific kinase complex (IKK) with small molecule drugs was reached in 1997. Frank Mercurio and co-workers from Signal Pharmaceuticals succeeded in the purification and cloning of an IKK signalosome containing an NF-κB specific kinase [54]. Signal Pharmaceuticals obtained a patent on the IKK signalosome with a priority date of 1997. Thus, it might not be surprising that Michael Karin, co-founder of Signal Pharmaceuticals was quite optimistic after the discovery of the IKK complex, that this might be an optimal drug target in oncology [32]. In his review, thalidomide is promoted, an old drug, which was developed further by Celgene where Michael Karin is at the Scientific Advisory Board of the Signal Research Division. Furthermore, long-used pain killers such as aspirin and sodium salicylate are known as efficient NF-κB-inhibitors at a concentration of 5 mM [55]. However, these pain killers were of no use in oncology [56]. These days, many of the major drug companies succeeded in the development of highly specific drugs for the inhibition of IKK2. An example is the pyridine derivative compound A (7-[2-(cyclopropylmethoxy)-6-hydroxyphenyl]-5-[(3S)-3-piperidinyl]-1,4-dihydro-2H-pyrido[2,3-d][1,3]oxazin-2-one hydrochloride) from Bayer Health Care AG (Wuppertal, Germany), which was reported to inhibit recombinant IKK-2 with Ki of 2 nM [57]. However, all of these specific IKK-2 inhibitors did not make their way in the clinics. Only thalidomide succeeded in oncology as an NF-κB inhibitor. Thalidomide is a long-known drug initially synthesized by the pharmaceutical company CIBA (Switzerland) in 1954 and used as a tranquilizer, sedative, and antiemetic for morning sickness. Unfortunately, Contergan (Thalidomide from Grünenthal, Aachen) taken during pregnancy had some unfortunate teratogenic side effects. Now the Grünenthal Foundation is one way to support thalidomide-affected people (https://www.grunenthal-foundation.com, accessed on 20 December 2021). Celgene now bought by Bristol Myers Squibb has already successfully exploited further commercial applications of thalidomide. These efforts lead to the development of thalidomide and a structural derivative: lenalidomide with no keto- but an additional amino group. Lenalidomide and pomalidomide (imnovid) are used as a treatment for various hematologic neoplasia such as multiple myeloma, myelodysplastic syndromes, and mantle cell. According to the European Medicines Agencies (https://www.ema.europa.eu/en/medicines/human/EPAR/revlimid, accessed on 20 December 2021), lenalidomide is a very effective drug and integrated into many therapeutic protocols used for multiple myeloma. It is used in combination with dexamethasone, or bortezomib and dexamethasone, or melphalan and prednisone. Surprisingly, NF-κB-activation was described in lung epithelial cells upon treatment with melphalan by Osterlund and coworkers [58].

In Germany, the Deutsche Gesellschaft für Hämatologie und Medizinische Onkologie e. V. has provided guidelines for integrative therapy for hematological disorders and solid tumors called onkopedia guidelines (https://www.onkopedia-guidelines.info/en/onkopedia/guidelines, accessed on 20 December 2021). The discussion of the treatment below is in accordance with these guidelines. More recently, chemotherapeutic schemes with lenalidomide were combined with immunotherapy utilizing daratumumab (anti-CD38 antibody) or elotuzumab (anti-SLAMF7 antibody). In myelodysplastic syndromes, a group of bone marrow disorders that cause anemia, lenalidomide is used in patients who need blood transfusions to manage their anemia, but only in patients with deletion of chromosome 5q, when other treatments are not adequate. In mantle cell lymphoma blood cancers that affect B lymphocytes, lenalidomide being used in adults is approved for patients with recurrent disease in combination with the antibody rituximab (anti-CD20). Thalidomide has anti-inflammatory, immuno-modulatory, and anti-angiogenic effects [59]. Thalidomide might act as a prodrug and its metabolites formed by oxidation via CYP2C19 might inhibit pro-inflammatory activation of the IKK complex, which could regulate genes for metastasis, proliferation, inflammation, angiogenesis, and anti-apoptosis. Interestingly, in tumor immunology thalidomide stimulated natural killer cells (NK-cells) in multiple myelomas accompanied by elevated levels of T-cell proliferation in an interleukin-2-dependent manner and an increase of γ-interferon production [60]. Furthermore, cytotoxicity of NK-cells was increased by thalidomide treatment in multiple myeloma [61].

Another strategy might be a screening of small molecules directly for cell death induction in CSCs. Following this direction, it was recently published, that CSCs from breast tumors could be killed far more effectively (100-fold) with the small molecule salinomycin compared to paclitaxel, which is a common drug for chemotherapy of breast cancer. Salinomycin is a compound whose action on CSCs was discovered by a chemical screen of more than 10,000 substances [62]. Salinomycin is especially active in CD44^high^/CD24^low^ cell lines suggesting a selective action on CSC-enriched subpopulations. There is some published pre-clinical evidence for salinomycin targeting NF-κB [44,63,64,65,66]. Furthermore, salinomycin and another natural compound: curcumin, making a yellow color in curry, were used as a magic bullet to target stem-cell specific receptor CD44 with a synthetic ligand [67]. Another interesting strategy to identify NF-κB-inhibiting substances by screening clinically approved small molecules was used by the NIH Chemical Genomics Center [68]. One out of 19 substances with high potency is the natural substance ectinascidin 743, which could inhibit IL1-induced NF-κB activation with a surprisingly low IC50 of 20 nM in living cells. Ectinascidin 743 (trabectedin) is used within the EU as Yondelis for the treatment of ovarian cancer and advanced soft-tissue sarcoma. In this line, trabectidin was reported to likewise induce efficient killing of CSCs in vitro [69]. Another chemotherapeutic agent inhibiting NF-κB is carboplatin [70].

Furthermore, other drugs used for the treatment of hematological diseases such as multiple myeloma in combination with other drugs are known NF-κB inhibitors such as the agonistic derivative of glucocorticoid hormones (corticosteroids) prednisone, or dexamethasone. There are two modes of action known: Direct inhibition of RelA by hormone-activated glucocorticoid receptor [71] and/or induction of the expression of IκB by glucocorticoid receptor [72]. The third medication used in combined chemotherapy is bortezomib, the first approved proteasome inhibitor. In our days, other proteasome inhibitors such as carfilzomib and ixazomib were developed. Bortezomib (Velcade9, developed by Millenium Pharamceuticals [73]) is widely used in multiple myeloma. One mode of action is the inhibition of proteasomal degradation of IκB [74].

### 3.3. Targeted Therapy of Cancer Stem Cells: An Excursion

The US National Cancer Institute defines targeted therapy as therapy that targets proteins that control how cancer cells grow, divide, and spread. Other ways of targeted therapy might include targeting specific cell types such as cancer stem cells by using biochemical characteristics of such cell types. Targeting specific proteins is one way to develop small molecule drugs such as tyrosine kinase inhibitors, as discussed above. Another way is Paul Ehrlich’s immunotherapy in new flavors with monoclonal and recombinant antibodies (see Figure 3). In the line of biochemical targeting, significant progress has been made for the treatment of CSCs when ferroptosis was discovered [75]. Ferroptosis is an iron-ion-dependent form of non-apoptotic cell death leading to oxidative cell death. Surprisingly, CSCs are accumulating iron [76] and could be specifically targeted with iron-chelating substances such as salinomycin. In this line, it was shown that salinomycin could induce iron load in lysosomes followed by iron-mediated production of reactive oxygen species leading to cell death [77]. Next to ferroptosis, approaches utilizing chimeric antigen receptor (CAR) T cells are increasingly recognized to directly target cancer [78,79] and cancer stem cells [80]. For instance, several studies showed the successful generation of anti-CD133, anti-EPCAM, or even anti-EGFR CAR T cells targeting CSCs from different parental tumors including glioblastoma or non-small-cell lung cancer [81,82,83] (see [80] for detailed review). Particularly targeting EGFR-signaling in glioblastomas via CAR T cells may prove a promising alternative to small molecules inhibiting EGFR as discussed above (see chapter 3.2.). Additionally targeting surface molecules with splice variants expressed on CSCs is a preclinical research strategy as in the case of anti-CD44 variant 6 CAR-T cells [84].

### 3.4. Anti-Inflammatory Cues Acting on NF-κB in Cancer Stem Cells

Activation of NF-κB in CSCs has been summarized in Figure 3. Here, we depict inhibitory cues acting on intracellular NF-κB (Figure 4). NF-κB could be activated by cancer-produced IL-1 (Figure 4, left cell). Inhibition of NF-κB-activation could be mediated by several members of the IκB family (for review see [85]). For an in-depth review of 750 inhibitors including intracellular proteins, we refer to [86]. These data are always updated on T. Gilmore’s webpage (https://www.bu.edu/nf-kb/gene-resources/target-genes/, accessed on 20 December 2021). There are many regulatory loops involving de-ubiquitinylating enzymes, which target signaling molecules for degradation (for review see [45]). Furthermore, non-coding RNAs such as microRNA (miR) 146 [87] or long non-coding RNAs NKILA [88] could inhibit NF-κB-activation by targeting activating proteins. Inhibition of transactivation could be driven by IEX-1 or by COMMD1-mediated ubiquitination of RELA (for review see [45]. As depicted on the right side of Figure 4, naïve CD4^+^ cells infiltrating the tumor can differentiate into induced regulatory T cells (iTregs). These express ectoenzymes such as CD39 [89] and CD73 responsible for the hydrolysis of ATP to adenosine [90], in turn inhibiting NF-κB-activation in CSCs. iTREGs could fancy cREL for expression of FOXP3 lineage transcription factor [91]. A pro-tumor function of iTREGs might be the repression of CD8 cytotoxic lymphocytes (CTL).

## 4. Conclusions: Lessons Learned for Cancer Stem Cell Therapy

This magic bullet strategy according to Paul Ehrlich might enhance specificity for CSCs and thus reduce systemic side effects. Surface markers for hematological CSCs, such as CD19, CD20, CD33, and CD38, are a good way to target CSCs for additional chemotherapy, e.g., with thalidomide analogs [92] (see also chapter 3.3). Limitations are an unclear correlation of these markers with CSCs. At least CD33 was shown to be on CSCs [93] and CD38 seems to be a biomarker for lung CSCs in mice [93], CD20 is likewise discussed to regulate cell growth and differentiation in humans via initiating intracellular signaling, but deletion of CD20 in mice did not result in any obvious phenotypes [94]. However, the role of CD20 as a bona fide marker of CSCs or just of plasma cells is still a matter of debate.

Taken together, until now no specific inhibitor of NF-κB has made it to the clinical applications. There might be several reasons for this: Baud and Karin suggest that “a molecularly targeted therapy should prevent NF-κB-activation without any effects on other signaling pathways, and be more active in malignant cells than in normal cells” [95]. While this seems to be a logical strategy, data from successful drugs like thalidomide, lenalidomide, and proteasome inhibitors might argue against a stringent selection strategy focusing solely on monospecific NF-κB-inhibitors. Successful drugs have a broad action on different targets, e.g., all substances degraded in the proteasome and at a first glance no special targets in cancer cells or CSCs. However, one might have to take into account that these drugs are only effective in liquid tumors such as multiple myeloma. It seems to be completely unclear why the same NF-κB inhibitory therapy does not work in solid tumors. In this line, a clinically used therapy on NF-κB downstream targets uses inhibition of IL1beta in rheumatoid arthritis with recombinant IL-1 receptor antagonists such as anakinra [96]. Other biologicals such as a monoclonal antibody against *IL1b* (canakinumab) might restore cytotoxic T-cell function in tumors, thus reducing pro-tumor inflammation. Canakinumab is currently developed by Novartis in a clinical trial (phase 3 study). *IL1b* negativity is a prognostic survival factor in breast tumors [97]. In humanized mouse NOD/SCID model with breast cancer cells, Anakinra or canakinumab reduced metastasis [98]. Interestingly, one of the most successful immunotherapies against solid cancers is the use of checkpoint inhibitors targeting PDL1-signalling (Figure 3). A direct transcriptional target of NF-κB in cancer cells is PD-L1 [99]. As a limitation, up to 50% of patients with tumors positive for PD-L1 display resistance or relapse despite being treated by blocking PD-1/PD-L1 [100]. After initially responding to the blockade of PD-1/PD-L1, most patients show resistance [101]. We propose a correlation of this phenomenon with NF-κB signaling.

## Figures and Tables

**Figure 1 biomedicines-10-00261-f001:**
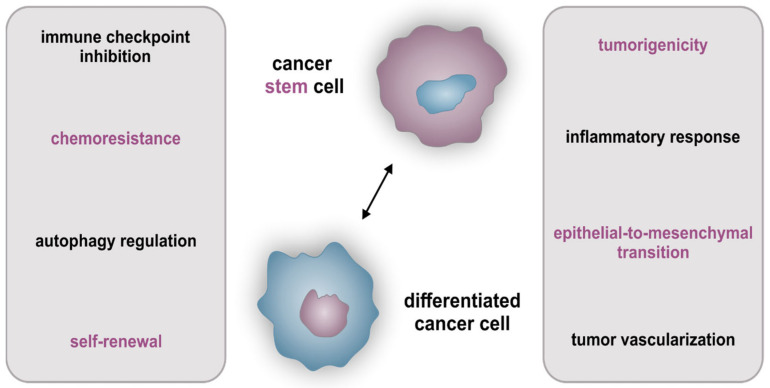
CSC- as well as incorporated cancer cell characteristics. Features and benefits for survival/maintenance of CSCs and (partially) differentiated cancer cells, whereas magenta characteristics exclusively describe cells possessing the stemness character.

**Figure 2 biomedicines-10-00261-f002:**
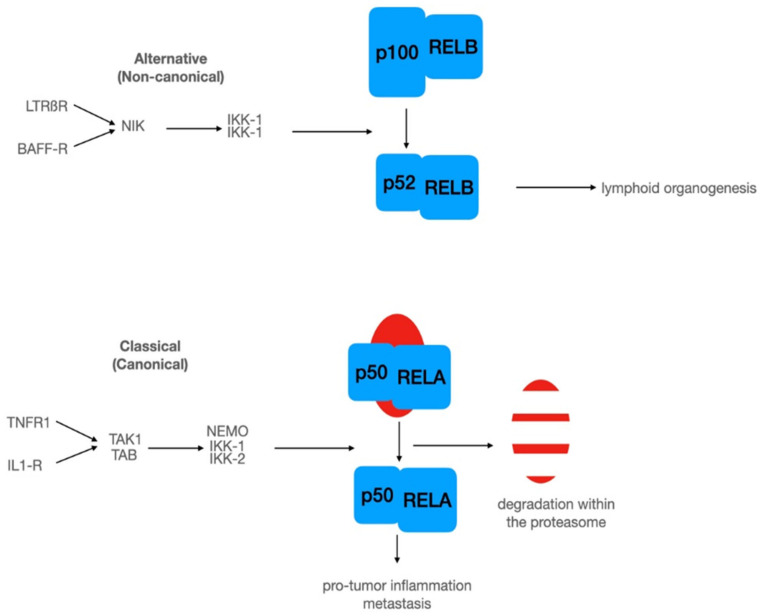
NF-κB signaling involves at least two different pathways: An alternative pathway with upstream kinase NIK and a dimer of IKK-1. In this pathway, the precursor of p52 acts as an inhibitory subunit. The classical pathway involves a dimer of upstream kinases (TAK1, TAB), which could activate a trimeric signalosome composed of NEMO (docking component), IKK-1, and IKK-2. Activation of signalosome leads to phosphorylation of IκB-α at serine 32 and 36, followed by ubiquitination and subsequent degradation within the proteasome. For further details, see the text.

**Figure 3 biomedicines-10-00261-f003:**
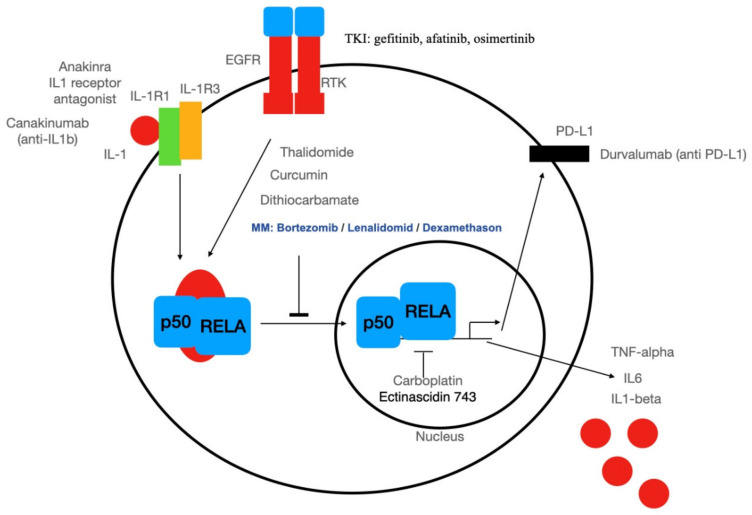
Overview of biologicals (anti-IL1β, anti-PDL-1) and small molecules such as RTK-inhibitors (RKI) for targeting NF-κB-signaling in cancer and CSCs.

**Figure 4 biomedicines-10-00261-f004:**
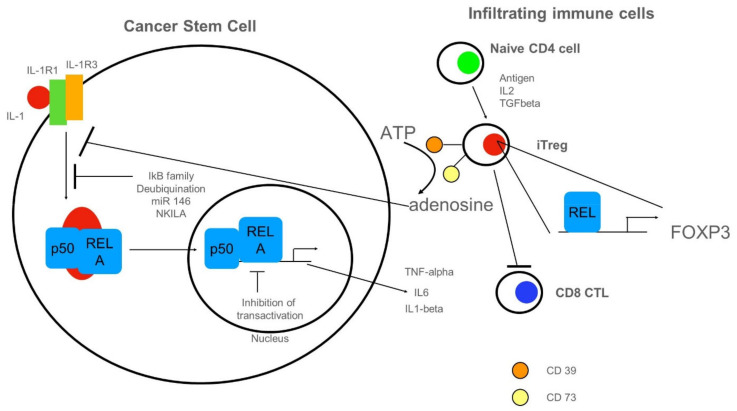
Anti-inflammatory cues impinging on NF-κB in CSCs. On the left, a CSC with NF-κB-signaling is depicted. NF-κB could be activated by cancer-produced IL-1. Inhibition of NF-κB-activation could be mediated by several members of the IκB family, by the de-ubiquitination of signaling molecules (for details see text). Furthermore, non-coding RNAs such as microRNA(miR) 146 or long non-coding RNAs NKILA could inhibit NF-κB-activation by targeting activating proteins. Inhibition of transactivation could be driven by IEX-1 or by COMMD1-mediated ubiquitination of RELA. On the right side, tumor-infiltrating immune cells are shown. Naïve CD4^+^ cells can differentiate into induced regulatory T cells (iTregs). These express ectoenzymes such as CD39 and CD73 responsible for the hydrolysis of ATP to adenosine, in turn, inhibiting NF-κB-activation in CSCs. iTREGs could fancy cREL for expression of FOXP3 lineage transcription factor. A pro-tumor function of iTREGs could be the repression of CD8 cytotoxic lymphocytes (CTL).

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
