# Peer review of "Targeting NF-κB Signaling in Cancer Stem Cells: A Narrative Review"

_biomedicines, 2022, doi:10.3390/biomedicines10020261_

Round 1
Reviewer 1 Report
The manuscript entitled "Targeting NF-κB signalling in cancer stem cells: a narrative review " summarizes the relationship between cancer stem cells and tumor development, the mechanism of NF-KB signaling, the strategy of targeting NF-κB, and the potential benefits of targeting NF-κB in cancer stem cells. The review has a potential impact. However, there is an issue that could be improved.
In line 181, Point 3 is a comprehensive summary for NF-κB, but there is only one aspect of 3.1, which seems inappropriate. It is recommended to list Point 4 in line 230 for “Systemic Therapy targeting NF-κB ”as 3.2.
Author Response
We thank the reviewer for this helpful remark. We now list “Systemic Therapy targeting NF-κB”as 3.2, as recommended.
Reviewer 2 Report
The manuscript by Kaltschmidt et al. is a very valuable work on NFkB signaling and its role in cancer stem cells. The review is very informative and up to date, it is well written and comprehensive, and a valuable reference for the readers interested in understanding the role of NFkB in cancer and cancer stem cells. The figures are well correlated to the main aspects described in the review.
- Line 20 Abstract: “we current strategies….” The verb is missing
- I suggest to include another figure summarizing all known inhibitors of NFkB pathway, not only drugs
- Another suggestion would be to include a short section on other types of therapies targeting NFkB; the focus in this review is on systemic therapy, but in contrast a short section on targeted therapies should be included, as a perspective of treatment.
